# Physiological and Transcriptome Analysis Reveals the Differences in Genes of Antioxidative Defense Components and Cold-Related Proteins in Winter and Spring Wheat during Cold Acclimation

Xiaoguang Lu [1], Yuhan Wu [1], Chaoyue Tang [1], Chang Liu [1], Ninghui Li [1], Yuchen Du [1], Lianshuang Fu [1], Xin Liu [1], Jun Liu [1,2] 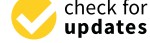 and Xiaonan Wang [1,*] 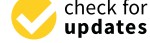

[1] College of Agriculture, Northeast Agricultural University, Harbin 150031, China
[2] National Key Facility for Crop Resources and Genetic Improvement, Institute of Crop Science, Chinese Academy of Agricultural Sciences, Beijing 100081, China
* Correspondence: xnwang1982@neau.edu.cn

**Abstract:** Recent findings suggest that cold acclimation can enhance cold resistance in wheat. Dongnongdongmai 1 (DM1) is a winter wheat variety that can overwinter at −30 °C; however, its cold acclimation mechanism is yet to be fully elucidated. Here, we elucidated the potential mechanisms of cold acclimation in DM1 and the China Spring (CS) variety, especially the role of the antioxidant system, using transcriptome and physiological analyses. Cold stress increased $H_2O_2$ and $O_2^-$ production in both varieties; however, CS had higher contents of $H_2O_2$ and $O_2^-$ than DM1. Moreover, cold significantly increased ROS-scavenging activities in DM1, especially at 30 days after exposure. Gene ontology (GO) analysis showed that differentially expressed peroxidase (POD) genes were enriched in antioxidant activity, with most POD genes being significantly upregulated in DM1 under cold acclimation. Additionally, cold acclimation increased the expression of cold acclimation protein (CAP), late embryogenesis abundant protein (LEA), and cold-responsive genes in both varieties, with higher expression levels in DM1. Overall, the results showed that DM1 exhibited a higher cold tolerance than CS during cold acclimation by increasing the expression of POD genes, LEA, CAP, and cold-responsive proteins, improving the understanding of the mechanism of cold resistance in DM1.

**Keywords:** cold acclimation; reactive oxygen scavengers; POD; cold tolerance genes



## 1. Introduction

Global warming provides conditions that may allow winter wheat to be farmed in areas close to the Arctic Circle. However, lack of winter hardiness may be a major environmental factor limiting winter wheat (*Triticum aestivum L.*) expansion into northern latitudes [1]. Low temperature limits crop growth and productivity by causing tissue damage through the formation of ice crystals [2,3], altering physiological processes [4], and inhibiting enzymatic activity [5–7]. Moreover, cold stress can induce the production of reactive oxygen species (ROS), which can cause damage to cell membranes [2]. Cold acclimation (exposure to low but nonfreezing temperatures) can enhance cold resistance in wheat [8]. During cold acclimation, wheat adapts to low temperatures through several processes, including the activation of cold-related genes [9], accumulation of osmoprotectants [10], production of antifreeze [11] or protective proteins [12], reorganization and rearrangement of the actin cytoskeleton [13], accumulation of polyunsaturated lipid [3], and increasing ROS scavenging [14].

Low temperature causes the overproduction of ROS in plants by affecting membrane fluidity, inhibiting photosynthesis, damaging nucleic acids and proteins, and altering carbohydrate metabolism, resulting in cell dysfunction and death [15,16]. The chloroplast is the

prime site for ROS production, producing 30–100 times more ROS than the mitochondria. Exposure to light excites chlorophyll (chl) pigments to a triple state, 3Chl*, and charge recombination may occur if the 3Chl* state is not efficiently quenched, inducing the excitation of $^3O_2$ to $O_2^-$ [17]. Moreover, $O_2^-$ can be produced through the Mehler reaction and later converted into $H_2O_2$ by superoxide dismutase (SOD) [16]. Although $H_2O_2$ and $O_2^-$ are the main ROS species, OH, $HO_2^-$, and RO are also produced during stress [18]. ROS enhances the expression of ROS-scavenging enzymes, including SOD, ascorbate peroxidase (APX), peroxidase (POD), catalase (CAT), and glutathione reductase (GR) [19]. Moreover, the expression of antioxidant enzymes is positively correlated with higher abiotic stress tolerance [20]; however, studies on the response of ROS-scavenger genes in wheat to cold acclimation are limited [21].

Wheat (*Triticum aestivum L.*) is a highly consumed staple crop worldwide and plays an important role in food security [22]. In the past few decades, cold stress during spring has been reported to cause severe losses to wheat production and grain quality [23]. Dongnong-dongmai 1 (DM1) is a winter wheat variety with high cold resistance and can survive in the cold regions of China (north latitude, 44–47°), where the average temperature in the coldest month is −10 to −8 °C [24]. Several studies have identified cold resistance genes in DM1, including *TaBADH* [25], *TaTPS11* [26], *TaG6PDH* and *Ta6PDH* [27], *TaFBA-A10* [28], and *MYC2* [29]. Additionally, long non-coding RNAs involved in cold stress response have been identified in DM1 [30]. Moreover, hormones [31] and the developmental stage of the wheat shoot apex have been shown to affect cold resistance in DM1 [32].

Omics technology offers the opportunity to study the regulatory mechanisms of complex networks related to cold hardiness. Lv et al. [33] performed a transcriptome analysis of wild-type wheat (Jimai325) and cold-sensitive mutants and identified eight upregulated enzymes in Jimai325, all of which were involved in the sucrose and amino acid biosynthesis pathway. Additionally, Tian et al. [34] reported that differentially expressed genes (DEGs) in DM1 exposed to temperatures ranging from −25 to 0 °C were involved in oxidation–reduction, protein phosphorylation, and carbohydrate metabolism. However, studies are yet to elucidate the role of the antioxidant system in cold tolerance in DM1, especially enzymes and genes involved in ROS scavenging under cold stress.

Therefore, the aim of this study was to elucidate the potential mechanisms of cold tolerance in wheat, especially the role of non-enzymatic and enzymatic antioxidants. To achieve this, we examined changes in the activity of ROS scavenging enzymes and the expression levels of related genes in winter wheat under cold stress conditions, using transcriptome and physiological analyses. It is anticipated that the findings of this study will improve the understanding of the physiological and molecular mechanisms of cold resistance in winter wheat.

## 2. Materials and Methods

### 2.1. Plant Materials

The winter wheat cv. Dongnongdongmai 1 (DM1) and the spring variety China Spring (CS) were used in this study. DM1 and CS have overwintering rates of >80% and <10%, respectively, in Haerbin Heilongjiang Province, China (N44°04′–N46°40′).

### 2.2. Plant Culture and Sampling

#### 2.2.1. Determination of Overwintering Rate

DM1 and CS were planted in a field at the experimental farm of the Northeast Agricultural University in Heilongjiang Province, China (45°44′26″ N; 126°43′41″ E), on 1 September in 2015, 2017, and 2018, and the overwintering rates of the varieties were determined the following spring. The experiment was laid in a randomized block design consisting of five replicates per group. Each genotype was planted in 3 m rows at a spacing of 20 cm between rows and 5 cm between plants. The re-greening rate was calculated as follows: number of surviving plants/number of plants before overwintering ×100%.

2.2.2. Plant Culture, Sampling of Physiological Indices, and RNA-seq

Healthy and full CS and DM1 seeds (30 each) were evenly planted in white plastic pots (35 cm long and 20 cm wide) in a light incubator (Jiangnan Instrument, RXZ-500B). The growth conditions of the incubator before the 3-leaf stage were as follows: temperature of 24 °C at 2000 lux for 16 h/18 °C at 0 lux for 8 h. The plants were randomly assigned to cold stress in an incubator (10 °C at 2000 lux for 16 h/4 °C at 0 lux for 8 h) or normal conditions (24 °C at 2000 lux for 16 h/18 °C at 0 lux for 8 h) after the 3-leaf stage.

Samples were collected from DM1 and CS at the 3-leaf stage, after the 3-leaf stage at 10 and 30 days under normal conditions, and after the 3-leaf stage at 10 and 30 days under cold conditions. A total of 30 plants were sampled at each time point, and the tillering nodes were quickly washed with clean water and cut. The 30 samples were divided into two groups: a group of 20 plants for physiological analysis and the other group of 10 plants for RNA-seq. The plants were snap-frozen in liquid nitrogen and stored at −80 °C.

*2.3. Determination of Electrical Conductivity, ROS CONTENT, and Antioxidants*

Electrical conductivity, $O_2^-$ content, and $H_2O_2$ concentration were determined according to the methods described by Elstner and Sagisaka [35,36]. Superoxide dismutase (SOD) was determined using the NBT reduction method [37], while peroxidase (POD) and catalase (CAT) activities were determined following the methods described by Wei [38] and Dou [39], respectively. Ascorbate peroxidase (APX) activity was assessed using the method of Nakano [40], while ascorbic acid (AsA) and glutathione (GSH) were determined using the method of Qin [41].

*2.4. RNA-Seq Analysis*

Total RNA was extracted using Trizol reagent (Invitrogen, Waltham, MA, USA) and treated with TURBO DNase I (Ambion, Waltham, MA, USA) for 30 min and then purified using RNeasy Plant Mini Kit (QIAGEN, Hilden, Germany). RNA sequence libraries were prepared using the TruSeq RNA sample Prep V2 kit (Illumina Inc., San Diego, CA, USA), according to manufacturer instructions. The quality and size of cDNA libraries for sequencing were verified using the Agilent 2200 TapeStation system (Agilent Inc., Santa Clara, CA, USA). RNA libraries were sequenced using the HiSeq 4000 sequencing system (Illumina Inc., San Diego, CA, USA) with a 150-cycle single-end sequencing protocol.

*2.5. Analysis of RNA-Seq Data*

The wheat genome sequence IWGSC RefSeq v1.0 (http://www.wheatgenome.org/News/Latest-news/RefSeq-v1.0-URGI accessed on 25 July 2022) and annotation files were downloaded from the Ensembl Plants database (http://plants.ensembl.org/Triticum_aestivum/Info/Index accessed on 10 October 2022). Hisat2 v2.1.0 was used to build index files for all the wheat chromosome and scaffold sequences. The RNA-seq reads were aligned against the bt2l-formatted Hisat2 index files using Hisat2 software. The relative expression levels of the genes were presented as fragments per kilobase of exon per million fragments mapped (FPKM) transcript and normalized using Cuffcompare, Cuffdiff, HTseq-count, and DESeq2 with global normalization parameters. Differential gene expression analysis was performed using Hisat2, SAMtools, Cufflinks, Cuffcompare, HTseq-count, and DESeq2. The read count of each gene was aligned and calculated using Hisat2, SAMtools, and HTseq-count based on a GTF-formatted file produced by Cufflinks and Cuffcompare. Subsequently, gene expression levels were normalized and differential gene expression analysis was performed using DESeq2 based on the negative binomial distribution (Anders and Huber 2010). Genes with normalized expression fold-change >2 at $p < 0.01$ and Benjamini–Hochberg adjusted $p$-value/false discovery rate (FDR) <0.1 were considered as differentially expressed genes (DEGs). A custom Perl script was used to integrate and summarize the results produced by Cuffdiff, HTseq-count, and DESeq2. The DEGs were grouped according to their expression levels.

### 2.6. Gene Ontology (GO) Analysis

GO enrichment analysis was performed using agriGO and GOEAST based on the hypergeometric test. Additionally, the data were aligned against the Arabidopsis genome to identify orthologs, using AGRIS and DATF databases. We used a custom Perl script to match the annotation information of Arabidopsis genes to wheat genes. Accession numbers, experimental information of RNA-seq data sets, and the fastaq-formatted sequences are available on the NCBI Gene Expression Omnibus database (http://www.ncbi.nlm.nih.gov/geo/ accessed on 10 October 2022) under IWGSC RefSeq annotation v1.0 downloaded from the Ensembl Plants database.

### 2.7. Quantitative RT-PCR

Total RNA was isolated from tillering nodes, using TRIzol reagent (Invitrogen, Carlsbad, MA, USA). First-strand cDNA was synthesized using EasyScript First-Strand cDNA Synthesis SuperMix (Transgen, Bejing, China), according to the manufacturer's instructions. The relative expression analysis was performed using a Roche LightCycler 480 system with TransStart Top Green qPCR SuperMix (Transgen, Bejing, China), as described by [26], with Arabidopsis actin 2 (ACT2) genes as the internal control. The relative expression of the target genes was calculated using the $2-\Delta\Delta Ct$ method (Livak and Schmittgen, 2001). The gene-specific primers used for qRT-PCR are listed in Table 1.

**Table 1.** Primers for real-time PCR.

| Gene ID | Forward Primer | Reverse Primer |
|---|---|---|
| *TraesCS2B01G614400* | CCGACCGTACGTACTGATTAAC | CGGTGAAGCCGACTGATTAT |
| *TraesCS6D01G054400* | GAGATTAGCAACCCTCCTTCTC | CTGGGCTCTCGAAGACATTT |
| *TraesCS6B01G063400* | CATCGGCTCACATCTTGTACTC | CGAACCAGATTAACGGCTCTTAT |
| *TraesCS7D01G347300* | GGCCTGAAATGGAGGAGAAA | CTACCCTGACGCAGATGTAAAG |
| *TraesCS3D01G158600* | CCAACCTCGTGCCCTTTAT | ATCGGGATCGCTGTCAAATTA |
| *TraesCS5B01G312000* | GTGGTGGTCAAGTACGCTAAT | CTCTCTTATAGCGGCAAGGATTT |
| *TraesCS1D01G411000* | TCATTCTCTCCGGTCCTACTT | AGTGCTGAAAGGCGAGATAC |
| *TraesCS2A01G048400* | CCTGCTGGTGAGGAAATTCA | CACCTTCTTCTTCTGTGCTCTC |

### 2.8. Analysis of the Evolutionary Tree of POD and Cis-acting Element

Annotated POD protein sequences (81) of Arabidopsis were downloaded from the TAIR database (http://www.arabidopsis.org/ accessed on 10 October 2022) and compared with POD proteins of wheat using Clustal W2. An evolutionary tree was constructed using the neighbor-joining method in MEGA7 software, with 500 bootstraps.

For cis-acting regulatory element analysis, 2 kb upstream sequences of *TaPOD* transcripts were downloaded from PlantCARE (http://bioinformatics.psb.ugent.be/webtools/plantcare/html/ accessed on 10 October 2022) and used for the prediction of cis-acting regulatory elements. The complete cis-acting element graphs were visualized using TBtools.

### 3. Results

#### 3.1. Comparison of Cold Resistance between DM1 and CS at Low Temperatures Stress

The freezing stage is defined as the period when the maximum daytime temperature is below 0 °C. In this study, the freezing stage started on 15 November 2019 (Figure 1D). DM1 and CS were sampled on 15 October (before freezing), 20 November (5 days after freezing), and 5 December (20 days after freezing). Plants in both groups had good growth before the freezing stage; however, some wilted leaves were observed in CS at 5 days after freezing, whereas the leaves of DM1 were unaffected. The above-ground leaves of CS were completely withered at 20 days after freezing, whereas only a few leaves were withered in DM1 (Figure 1A). Additionally, the regreening rates of DM1 in 2016, 2018, and 2019 were 100, 94, and 100%, respectively, whereas those of CS were 0, 1, and 0%, respectively (Figure 1B). The relative electrical conductivity of CS and DM1 was determined

on 10 October (before freezing) and 20 November (5 days after freezing) in 2018 and 2019. There was no significant difference in conductivity between DM1 and CS on 10 October of both years; however, CS had significantly higher conductivity than DM1 on 20 November (Figure 1C). In 2018, the conductivity of CS was 23.58% higher than that of DM1; in 2019, the conductivity of CS was 31.45% higher than that of DM1.

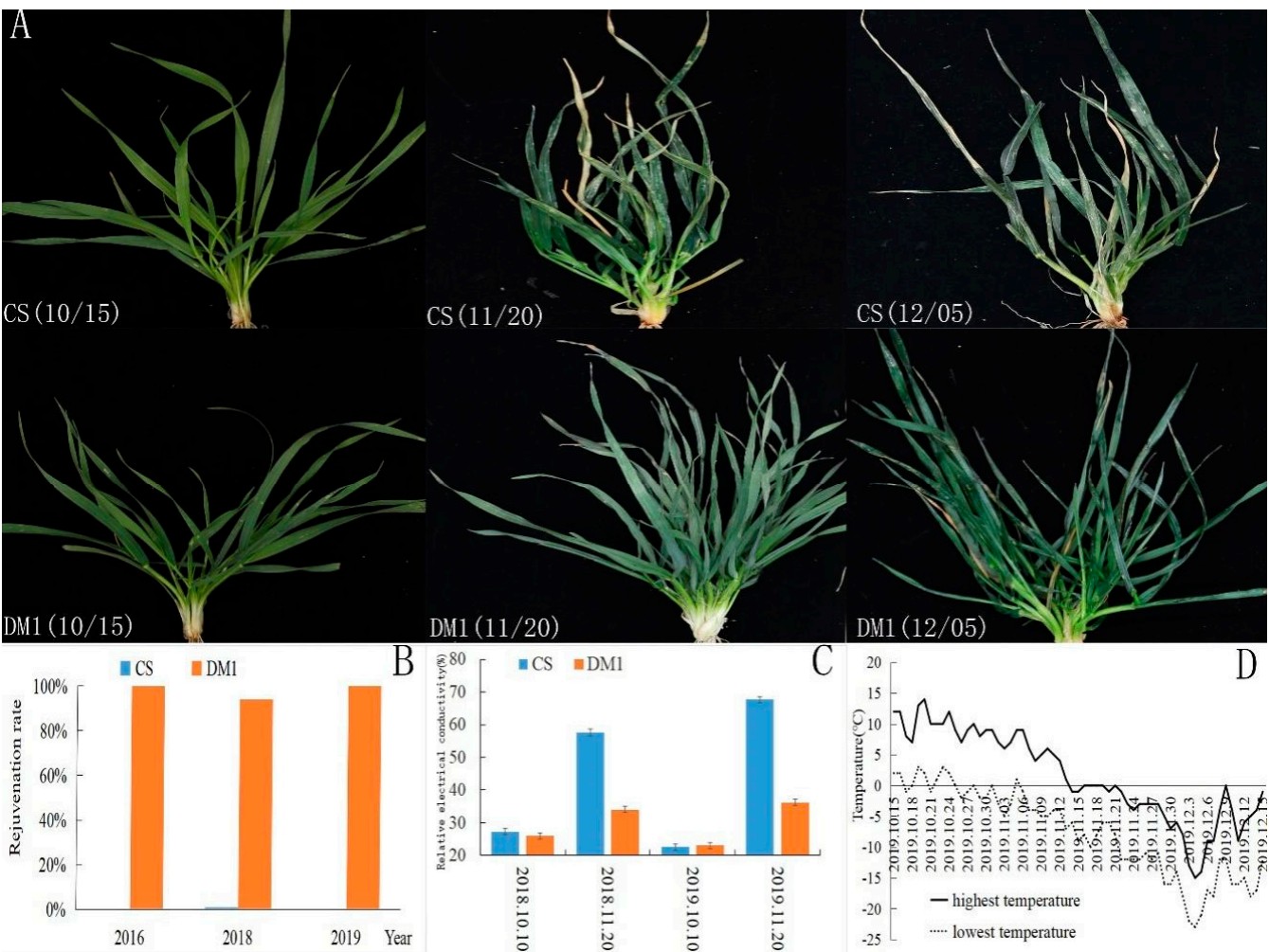

**Figure 1.** Differences in cold stress response between Dongnongdongmai 1 (DM1) and China Spring (CS): (**A**) phenotypes of DM1 and CS in the field on 15 October, 20 November, and 5 December in 2019; (**B**) field greening rates of DM1 and CS in 2016, 2018, and 2019; (**C**) relative electrical conductivity of the leaves of DM1 and CS in 2018 and 2019; (**D**) field meteorological data in 2019. The solid line represents the daily maximum temperature, and the dotted line represents the daily minimum temperature.

*3.2. Response of ROS and ROS-Scavenging Enzymes to Cold Acclimation in DM1 and CS*

The ROS contents of DM1 and CS wheat varieties were determined at 10 and 30 days after the three-leaf stage and exposure to cold or normal conditions to elucidate the effect of cold stress on ROS production (Figure 2). There were no significant differences in $H_2O_2$ and $O_2^-$ contents between the two varieties under normal conditions (24 °C). However, cold stress caused an increase in the $O_2^-$ contents of CS (0.44 and 0.68 nmol $g^{-1}$ FW $min^{-1}$) compared with those of DM1 (0.20 and 0.42 nmol $g^{-1}$ FW $min^{-1}$) after 10 and 30 days of cold acclimation, respectively (Figure 2A). Similarly, there was an increase in the $H_2O_2$ content of CS (7.68 μmol/g Fw) compared with that of DM1 (4.33 μmol/g Fw) after 30 days of cold acclimation (Figure 2B).

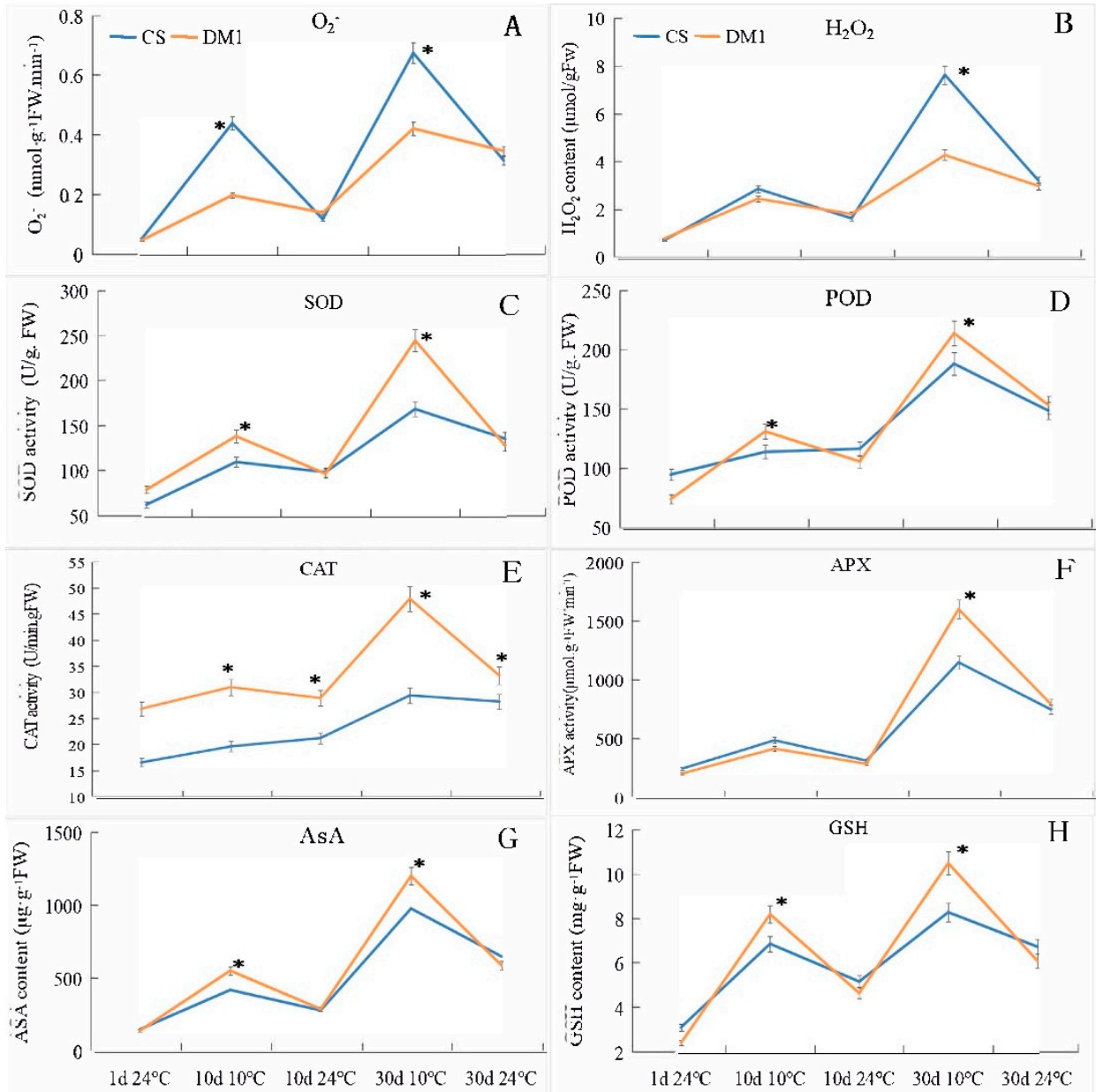

**Figure 2.** The concentrations of reactive oxygen species (ROS) and the activities of ROS scavengers in DM1 and CS: (**A**,**B**) the contents of $O_2^-$ and $H_2O_2$, respectively; (**C**–**F**) the activities of superoxide dismutase (SOD), peroxidase (POD), catalase (CAT), and ascorbate peroxidase (APX), respectively; (**G**,**H**) the contents of ascorbic acid (AsA) and glutathione reductase (GSH), respectively. Orange dash represents DM1, blue dash represents CS, * indicates significance at 5%.

Additionally, we examined the response of the antioxidant system to cold acclimation; particularly, the activities of SOD, POD, CAT, and APX and the contents of AsA and GSH were evaluated. There was a significant increase in the activities of SOD and the contents of AsA and GSH in both varieties after 10 and 30 days of cold acclimation (Figure 2C–H); however, the activities of the enzymes, except APX, were significantly higher in DM1 than in CS (Figure 2C–H). Specifically, the SOD activities of DM1 after 10 and 30 days of cold acclimation (139.4 and 245.8 U/g FW) were higher than those of CS (110.8 and 169.7 U/g FW) (Figure 2C). Similarly, the POD activities of DM1 and CS after 10 and 30 days of cold acclimation were 132.11 and 214.92 U/g FW (DM1) and 114.80 and 189.11 U/g FW (CS), respectively (Figure 2D). The CAT activities of DM1 and CS at 30 days of cold acclimation were 48.17 and 29.65 U/min g FW, respectively (Figure 2E). The APX activity of DM1

(426.04 µmol g$^{-1}$ FW-min$^{-1}$) was lower than that of CS (496.47 µmol g$^{-1}$ FW-min$^{-1}$) after 10 days of cold acclimation, but higher (1611.58 µmol g$^{-1}$ FW·min$^{-1}$) than that of CS (1159.87 µmol g$^{-1}$ FW-min$^{-1}$) after 30 days of cold acclimation (Figure 2F). The contents of AsA and GSH were significantly higher in DM1 (AsA: 1211.45 µg g$^{-1}$ FW and GSH: 10.55 mg g$^{-1}$ FW) than in CS (AsA: 987.16 µg g$^{-1}$ FW and GSH: 8.33 mg g$^{-1}$ FW) after 30 days of cold acclimation (Figure 2G,H). These results indicated that cold acclimation stimulated the synthesis of AsA and GSH in the cold-resistant variety DM1.

### 3.3. Transcriptome Profiles of the Tiller Nodes of CS and DM1

RNA sequencing of the tiller nodes of CS and DM1 was performed to identify changes in the expression levels of ROS-scavenging genes in response to cold stress. A total of 198 G of clean reads were obtained, with a Q30 value of 89.53%, among which approximately 92.2% of the reads were successfully mapped to two reference genomes (IWGSC 1.0 and Ensembl Plants genome), and 95.37% of the properly paired reads were uniquely matched. The exon structures of transcription units (TUs) were assembled using Cufflinks, SAMtools, HTseq-count, and DESeq2, and a total of 226,631 expressed TUs were identified, including 81,526 which overlapped with annotated genes and 27219 unannotated TUs or intergenic Tus (Figure S1A). At 10 and 30 days of cold acclimation 18,717 and 22,763 genes had significantly higher expression in CS than in DM1, while 19,688 and 17,025 genes had significantly higher expression in DM1 than in CS, respectively (Figure S2). Additionally, 8576 and 3584 TUs were specific to CS and DM1, respectively, and 62,241 TUs were detected in both genotypes (Figure S1B). The pathways in which DM1-specific expressed genes are involved are mostly developmental and involved in response to stimulus (Figure S1C).

The expression profiles of eight randomly selected genes were determined by qRT-PCR using specific primers (Table 1) to determine the reliability of the RNA-seq data. Regression analysis of the qRT-PCR and RNA-seq data showed that the lowest R2 value was 0.667 and the highest was 0.9204 for the eight randomly screened genes, indicating that the qRT-PCR data were consistent with the RNA-seq data and reliable (Figures 3 and S3).

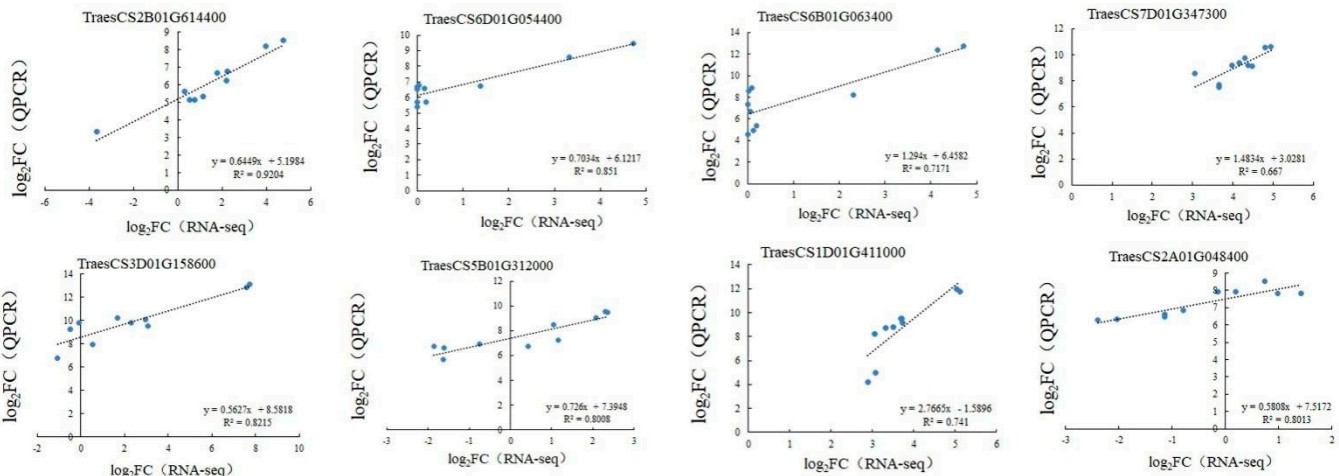

**Figure 3.** Comparison of RNA-seq and qRT-PCR data. The horizontal coordinate represents the log$_2$ FPKM value of a randomly selected gene and the vertical coordinate represents the log$_2$qPCR value of the same gene.

### 3.4. Expression Analysis of Antioxidant Genes in DM1 and CS

Differential expression analysis was performed to identify DEGs in DM1 and CS under cold acclimation, using DESeq2 software. A total of 82,248 DEGs were identified, including 58,802 annotated and 23,446 unannotated TUs (Figure 4A). Additionally, 1794 specific TUs were detected during cold acclimation, 6945 TUs were detected under normal conditions, and 27,173 TUs were detected under both treatments (Figure 4B). The DEGs were classified into 100 groups according to their expression levels (Table S1). There were considerable

variations in the number of DEGs in DM1 and CS before cold acclimation and after 10 and 30 days of cold acclimation (Figure 4C–J). Functional annotation of the DEGs showed that 158 POD, 119 GSH, 10 SOD, 8 CAT, and 3 APX genes were differentially expressed among the varieties or treatments (Table S2). Specifically, cold acclimation increased the expression of SOD genes; however, CS had higher expression levels of all SOD genes, except TraesCS4B01G243200 and TraesCS4D01G059500, than DM1 after 10 and 30 days of cold acclimation (Table S2, Figure 5A). Cold acclimation downregulated eight CAT genes; however, DMI had higher expression levels of the CAT genes than CS after 10 days of cold stress (Table S2, Figure 5A). Additionally, cold acclimation upregulated APX genes, with DM1 having a higher expression of the genes than CS (Table S2, Figure 5A). A total of 58 out of 119 GSH genes were upregulated by cold acclimation, with 47 and 43 DEGs significantly higher in DM1 than in CS after 10 and 30 days of cold acclimation, respectively (Table S2, Figure 5B). Moreover, 70 out of 158 POD genes were significantly upregulated after cold acclimation, with 40 and 37 POD genes having higher expression levels in DM1 than in CS after 10 days and 30 days of cold acclimation, respectively (Table S2, Figure 5C). GO functional annotation was performed on the genes that had significantly higher expression in DM1 than in CS, and peroxidase genes were found to be significantly enriched (Figure 6).

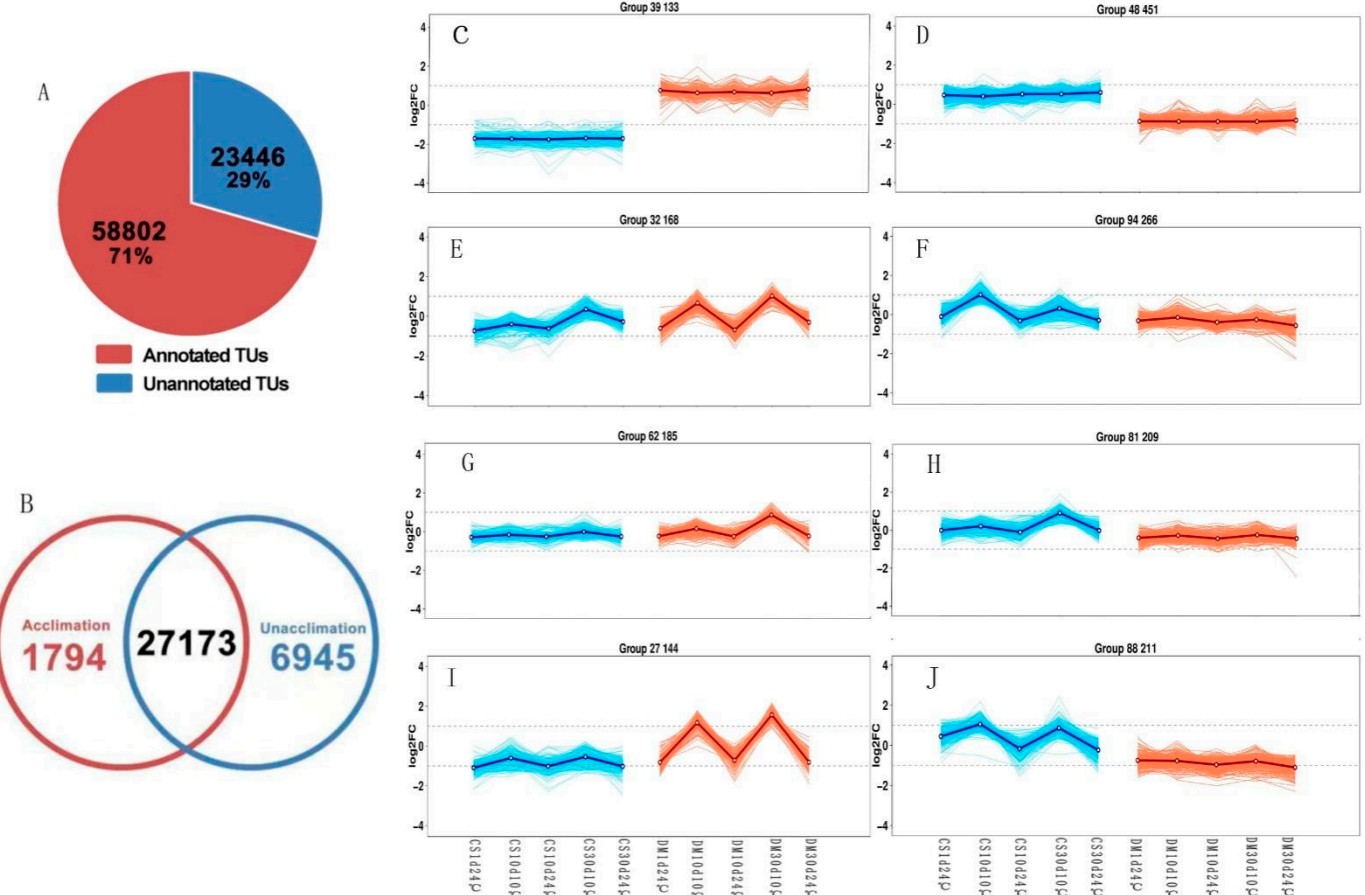

**Figure 4.** Differentially expressed genes (DEGs) in DM1 and CS: (**A**) DEGs in DM1 and CS were identified using RNA-seq analysis. (**B**) Venn diagram of DEGs under cold acclimation and normal temperature. (**C**–**J**) Clustering analysis of DEGs: (**C**,**D**) groups 39 and 48 indicate significantly upregulated and downregulated DEGs between DM1 and CS at all time points; (**E**,**F**) groups 32 and 94 indicate significantly upregulated and downregulated DEGs between DM1 and CS after 10 days of cold acclimation; (**G**,**H**) groups 62 and 81 indicate significantly upregulated and downregulated DEGs between DM1 and CS after 30 days of cold acclimation; (**I**,**J**) groups 27 and 88 indicate significantly upregulated and downregulated DEGs between DM1 and CS during cold acclimation.

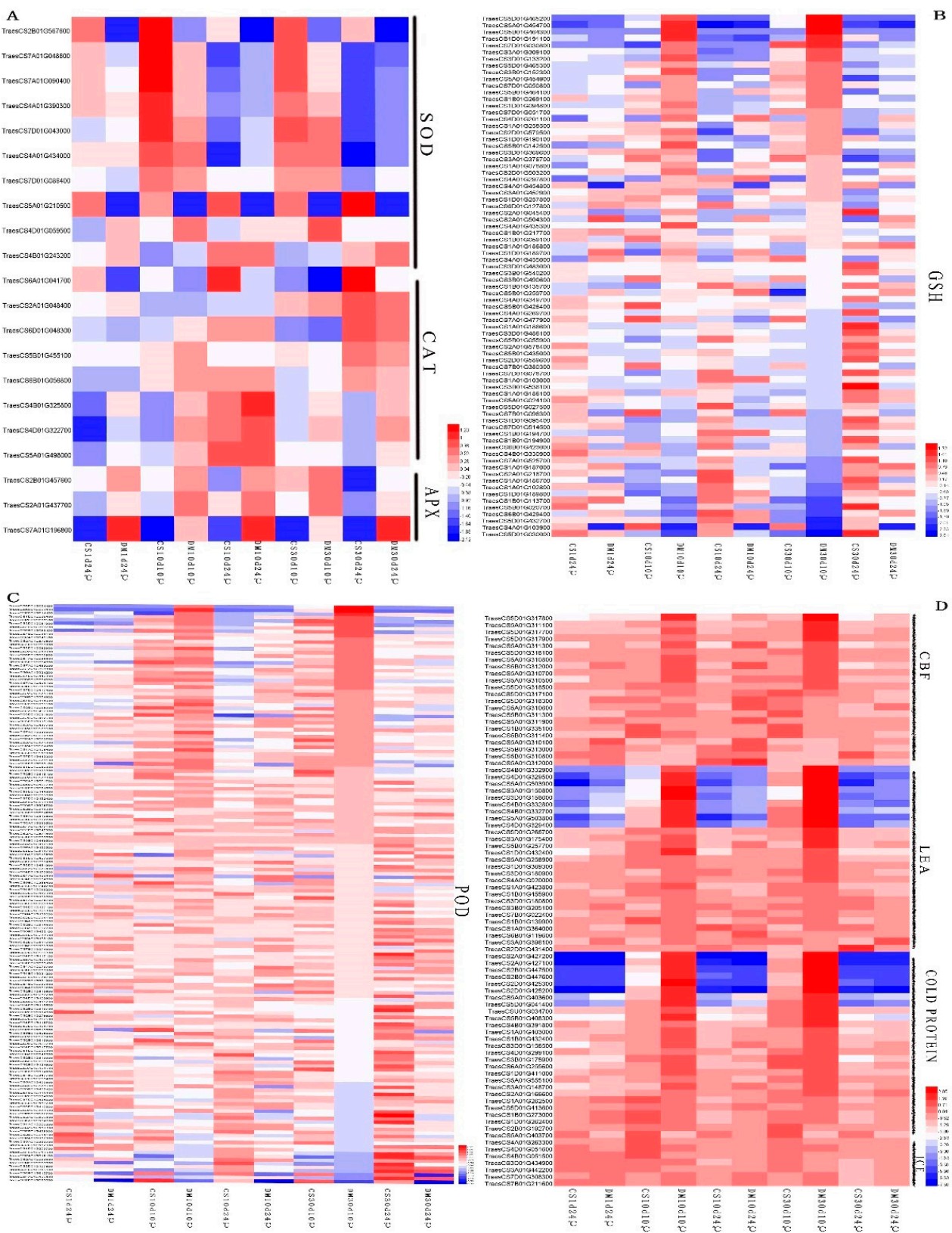

**Figure 5.** Heatmap of reactive oxygen species (ROS)-scavenging differentially expressed genes (DEGs) between CS and DM1: (**A**) SOD, CAT, and APX; (**B**) GSH; (**C**) POD; (**D**) CBF, ICE, LEA, and cold resistant protein.

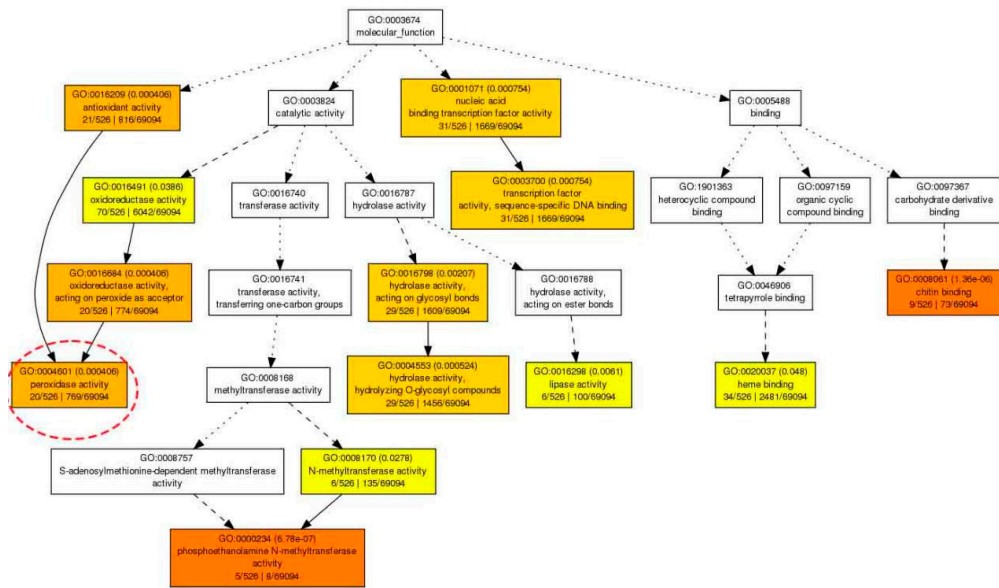

**Figure 6.** GO functional annotation of DEGs with higher expression in DM1 than in CS. * The red dotted line shows the peroxidase gene being significantly enriched.

Furthermore, a total of 35 POD genes were obtained after screening out genes with FPKM >10. Among the 35 POD genes, 23 and 24 had higher expression levels in DM1 than in CS after 10 and 30 days of cold acclimation, respectively. The amino acid sequences of the 35 POD genes were compared to those of Arabidopsis (https://www.arabidopsis.org/?tdsourcetag=s_pcqq_aiomsg accessed on 10 October 2022), and an evolutionary tree showing their relationship was constructed (Figure 7A). Among the POD genes, the expression levels of TraesCS7A01G339600, TraesCS6D01G303900, TraesCS6D01G054400, TraesCS6B01G063400, TraesCS2B01G614400, TraesCS7B01G251100, and TraesCS1B01G115900 were significantly higher in DM1 than in CS (Figure S4).

Cis-acting elements in promoters are crucial regions in the transcription factor binding site for initiating transcription and play a vital role in the regulation of gene expression. To further elucidate the possible biological functions of 35 TaPOD genes, we predicted the cis-acting regulatory elements in the 2 kb upstream promoter region using the online database PlantCare (Figure 7B). The promoter regions of wheat POD genes contain multiple cis-acting elements involved in light response, stress response, drought inducibility, low-temperature response, abscisic acid response, MeJA response, circadian regulation, and anaerobic induction. Overall, these results indicated that POD family members in wheat may play a role in phytohormone and stress responses. Additionally, the numbers of phytohormone response elements and stress response elements in the promoters of POD members differed considerably.

### 3.5. Expression of Cold Acclimation RESISTANCE Genes

Furthermore, the expression levels of cold shock protein, cold-responsive protein, c repeat binding factor (CBF), cold acclimation protein (CAP), and cold-regulated (COR) protein in the varieties were determined and illustrated using a heatmap (Figure 5D). The expression levels of COR protein, CBF, and inducer of CBF expression (ICE) were low. Cold acclimation did not significantly improve the expression of these genes, and the expression levels of some CBF and ICE genes were downregulated (Figures 5D, S5 and S6, Table S3). In contrast, cold acclimation significantly increased the expression of cold shock protein in CS and DM1 by 2.5–3.65-fold and 1.47–1.72-fold, respectively, after 10 and 30 days of cold acclimation, compared with the expression levels under normal conditions (Table S3). Similarly, cold acclimation significantly increased the expression levels of cold-induced proteins; however, DM1 had higher expression of cold-induced proteins than CS under all

temperature treatments (Table S3, Figure S7). Additionally, cold acclimation significantly increased the expression levels of cold acclimation proteins and cold-responsive proteins by 2–27-fold and 2069–3646-fold, respectively, after 10 days of cold acclimation compared with the expression levels under normal conditions. Moreover, DM1 had significantly higher expression levels of cold acclimation proteins and cold-responsive proteins than CS under cold acclimation (Tables 2 and S3, Figures S8 and S9). Cold acclimation significantly upregulated the expression levels of 17 late embryogenesis abundant (LEA) proteins in the varieties; however, DM1 had significantly higher expression levels of the genes than CS under cold acclimation (Table S3, Figures 5 and S10).

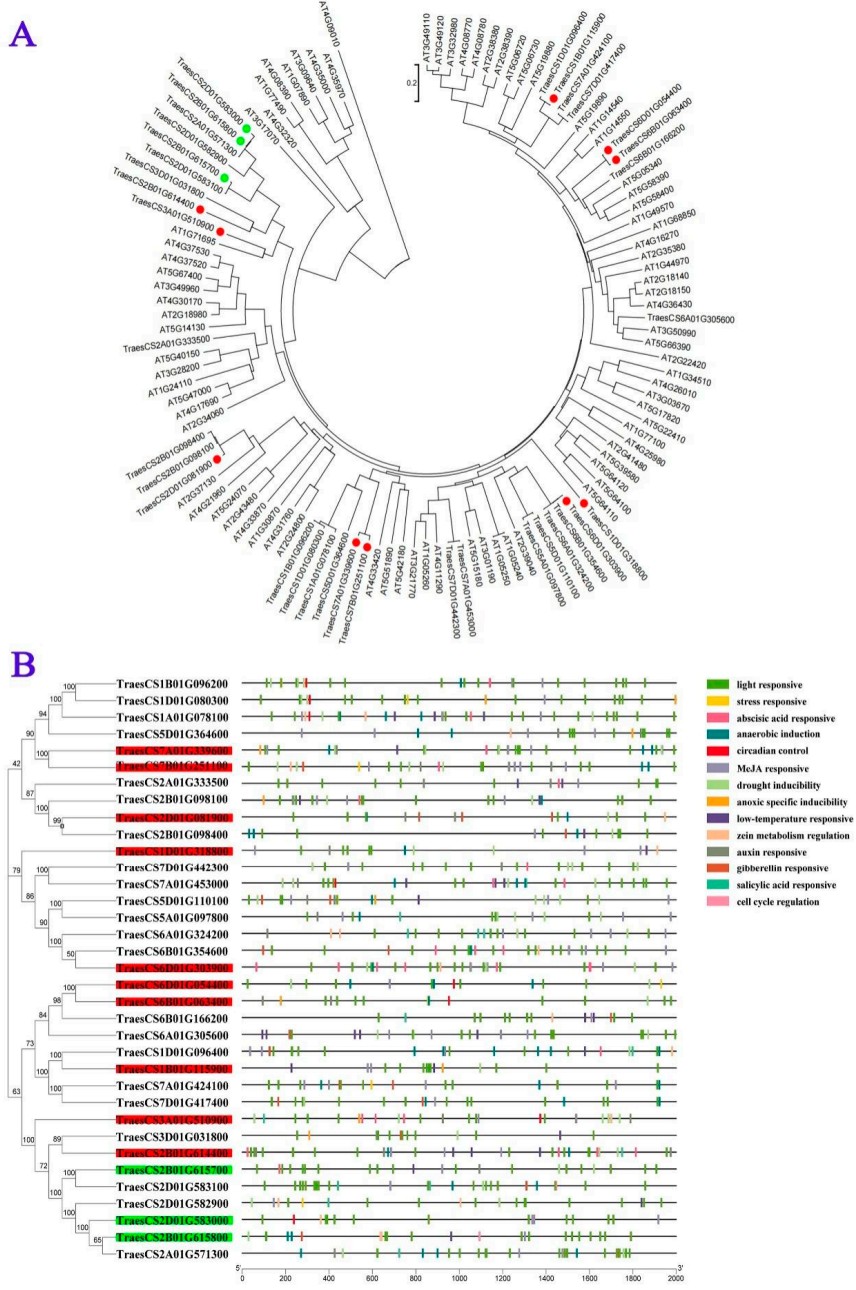

**Figure 7.** Evolutionary tree of POD genes and promoter cis-acting elements in wheat: (**A**) a total of 35 differentially expressed *TaPOD* and *AtPOD* genes were screened for evolutionary tree analysis; (**B**) analysis of promoter elements of 35 differentially expressed *TaPOD* genes. * Red represents significantly upregulated *TaPOD* in DM1 compared with CS, and green represents significantly downregulated *TaPOD* in DM1 compared with CS.

**Table 2.** Comparison of the expression levels of cold-related proteins between CS and DM1.

| Log2 Fold-Change | Gene ID | CS Log2 A/B | DM1 Log2 A/B | CS Log2 C/D | DM1 Log2 C/D |
|---|---|---|---|---|---|
| Cold-responsive protein | *TraesCS5A01G403600* | 6.65 | 11.51 | 7.80 | 10.86 |
| | *TraesCS5B01G408300* | 7.48 | 11.23 | 6.06 | 9.97 |
| | *TraesCS5D01G041400* | 8.00 | 11.23 | 7.29 | 10.93 |
| | *TraesCS1D01G411000* | 4.03 | 14.51 | 7.16 | 11.75 |
| | *TraesCS1B01G432400* | 5.93 | 11.83 | 6.46 | 11.12 |
| | *TraesCS1A01G403000* | 6.50 | 14.34 | 7.13 | 12.53 |
| | *TraesCS5A01G403700* | 0.50 | 0.89 | 1.11 | 1.50 |
| Cold acclimation protein | *TraesCS5A01G403600* | 2.73 | 3.78 | 3.53 | 4.56 |
| | *TraesCS5B01G408300* | 1.97 | 3.73 | 3.28 | 4.61 |
| | *TraesCS5D01G041400* | 4.13 | 4.78 | 6.01 | 6.48 |
| | *TraesCS1D01G411000* | 0.22 | 1.32 | 0.80 | 2.04 |
| | *TraesCS1B01G432400* | 0.76 | 1.52 | 1.35 | 2.19 |
| | *TraesCS1A01G403000* | 0.52 | 1.44 | 1.67 | 2.66 |
| | *TraesCS5A01G403700* | 0.54 | −0.75 | 0.56 | −0.35 |
| Cold-regulated protein | *TraesCS3D01G156500* | 0.5 | 0.89 | 0.15 | 0.59 |
| | *TraesCS3A01G148700* | −0.19 | 0.63 | −0.31 | 0.21 |
| | *TraesCS3D01G156500* | 0.5 | 0.89 | 0.15 | 0.59 |
| | *TraesCS3B01G175900* | 1.04 | 1.55 | 1.54 | 1.24 |

A: cold acclimation for 10 d after 3-leaf stage; B: normal conditions for 10 d after 3-leaf stage; C: cold acclimation for 30 d after 3-leaf stage; D: normal conditions for 30 d after 3-leaf stage.

## 4. Discussion

Over the years, plants have evolved intricate mechanisms to survive under stressful conditions. Studies have shown that several plant species exhibit decreased sensitivity to cold stress after undergoing prior exposure to low temperatures, a phenomenon called cold acclimation [42]. Cold acclimation causes molecular and physiological modifications in plants, including the accumulation of cytosolic $Ca_2^+$, increased levels of ROS, and increased activities of ROS-scavenging enzymes [43]. For instance, the exposure of spring oilseed rape to 3 days of cold acclimation temperature resulted in improved cold tolerance [44]. In the present study, ROS-scavenging enzymes and genes were involved in cold acclimation.

### 4.1. Changes in the Antioxidant System of Wheat under Cold Conditions

Low levels of ROS play a role in signal transduction, but an excessive amount of ROS can induce plant cell damage under stress [45,46]. Moreover, SOD can catalyze $O_2^-$ to form $H_2O_2$, which has a longer half-life and is more toxic to cells [47]. Moreover, Yu [48] reported that exposure to $-10$ °C temperature significantly increased the $H_2O_2$ content of the leaves and tillers of DM1. In this study, CS had significantly higher $H_2O_2$ and $O_2^-$ contents under cold acclimation than under normal conditions. There was no significant difference in $H_2O_2$ content between the varieties after 10 days of cold acclimation; in contrast, $H_2O_2$ and $O_2^-$ contents were significantly higher in CS than in DM1 at 30 days of cold acclimation (Figure 2), which was consistent with the findings of Quan [49]. These results showed that $O_2^-$ formation is more rapidly induced than $H_2O_2$ synthesis under low temperatures.

Several studies have confirmed the importance of the antioxidant system in plant protection against various stresses [50,51]. Cold acclimation and stress have been reported to significantly increase SOD, APX, and CAT activity and GSH content in chickpea [52]. In the present study, exposure to cold acclimation for 30 days significantly increased the contents of AsA and GSH and the activities of SOD, POD, CAT, and APX in both varieties of wheat; however, DM1 had significantly higher activities of the enzymes than CS. CAT, APX, and POD have been shown to play an essential role in $H_2O_2$ scavenging, thus protecting cells of higher plants against oxidative damage [47]. Jiang et al. showed that SOD, POD, and CAT activities increased with decreasing temperature, and the antioxidant enzyme activities of the cold-tolerant variety YN19 were all higher than those of the cold-sensitive

variety XM26, demonstrating that wheat has a tendency to mitigate low-temperature stress damage by enhancing the enzyme activities of the antioxidant system, and that cold-tolerant and cold-sensitive varieties have different scavenging efficiencies for ROS [53]. In this study, APX showed the highest increase among the six ROS-scavenging enzymes from 10 to 30 days after cold acclimation. APX activity increased by 278% in DM1 compared with that in CS, which increased by 134% (Figure 2). Additionally, there was no significant difference in CAT activity between DM1 and CS after 10 days of acclimation. Moreover, CS had a higher POD activity than DM1 at the three-leaf stage and 10 days after three-leaf stage under normal temperature; however, the opposite trend was observed under cold acclimation (Figure 2). The significantly higher activities of POD, CAT, and APX in DM1 than in CS after 30 days of cold acclimation could be responsible for the lower level of $H_2O_2$ in DM1 after 30 days of cold acclimation.

The effect of cold acclimation on the expression of ROS-scavenging genes was examined in the present study. Cold acclimation did not significantly increase the expression of CAT genes, but it significantly increased the expression of APX genes, including *TraesCS2A01G437700* and *TraesCS2B01G457600*, after 30 days of cold acclimation compared with normal temperature (Table S2). Additionally, GO functional annotation showed that POD genes were significantly enriched (Figure 6), among which 10 genes, including *TraesCS1B01G115900* and *TraesCS6D01G054400*, were significantly upregulated in DM1 than in CS under cold acclimation (Figure 7, marked by red dots), and were involved in cold acclimation response. Moreover, phylogenetic analysis showed that *TraesCS1B01G115900* was closely related to AT5G19880. Luo et al. [54] reported that the drought-resistant wheat variety SN20 had significantly higher expression levels of POD genes than the drought-susceptible variety F287 under drought conditions, and co-expression analysis showed that both wheat varieties exhibited opposite patterns of *TraesCS1B01G115900* expression. Additionally, *TraesCS6D01G054400* and *TraesCS6B01G063400* were closely related to *AT1G14550* (Figure 7A), and studies have shown that *AT1G14550* is involved in several abiotic stresses, including drought, hypoxia, and salt stress [55,56]. Moreover, *TraesCS1D01G318800* and *TraesCS6D01G303900* were closely related to *AT2G39040*. A previous study showed that *AT2G39040* expression is negatively correlated with $H_2O_2$ levels in sos6-1 mutant Arabidopsis exposed to salt stress [57]. *TraesCS2D01G081900* was closely related to *AT2G37130*, while *TraesCS2B01G614400* and *TraesCS3A01G510900* were closely related to *AT1G71695*. Wang et al. [58] found that ERF6 can bind specifically to ROSE7/GCC box (contains *AT2G37130*) and enhance the expression of a reporter gene. Bertini et al. [59] reported an increase in *AT1G71695* expression in *Colobanthus quitensis* under a 4 °C condition. In the present study, the high expression of the 10 POD genes (Figure 7, marked by red dots) in DM1 under cold acclimation may be related to the activity of POD.

*4.2. Response of Cold Resistance Genes in Wheat Varieties to Cold Acclimation*

Previous studies have identified some key genes involved in cold resistance in plants, including CAP, COR, CBF, ICE, C-repeat/dehydration-responsive (CRT/DRE), WCS, dehydrins (DHNs), and LEA [60]. CBF plays a crucial role in transcriptional pathways related to cold tolerance, and it can induce the expression of cold response genes by binding to ICE [28]. The *ICE-CBF-COR* signaling pathway has been shown to play important role in cold stress resistance in plants [61]. For instance, *LeCOR413* overexpression has been shown to reduce cell membrane damage, ROS accumulation, and PSII photoinhibition and enhance the activity of antioxidant enzymes and the content of osmotic regulators [9]. Ganeshan et al. [62] reported that the expression of COR 14 in near-isogenic lines (NILs) of winter wheat cultivar Norstar and spring wheat cultivar Manitou was in the order of NO (winter wheat) > WM (winter Manitou) > SN (spring Norstar) > MA (spring wheat) after 2 days of exposure to cold stress. In the present study, CBF and ICE had low expression levels in both DM1 and CS, and there were no significant differences in expression levels between DM1 and CS under the different temperature conditions (Figures S5 and S6). Similarly, the expression of COR protein was extremely low (Table S3). Therefore, we speculated that

CBF, ICE, and COR may be short-induced. Moreover, the plants were sampled after 10 and 30 days of cold acclimation in the present study. In contrast, the expression levels of CAP and cold-responsive genes were significantly upregulated by cold stress in both DM1 and CS; however, DM1 had significantly higher expression levels of the genes (4.87–8.87-fold higher) than CS (Table S3, Figures S8 and S9). A previous study showed that cold tolerance was significantly enhanced in Populus by transforming it with CAP from *Tamarix hispida* (*ThCAP*) [63]. Notably, six cold-responsive genes, namely *TraesCS2D01G425200*, *TraesCS2A01G427200*, *TraesCS2A01G427100*, *TraesCS2B01G447600*, *TraesCS2D01G425300*, and *TraesCS2B01G447500*, were significantly upregulated in response to cold acclimation (Figure S9, Table S3). Kippes et al. [64] reported that *TraesCS2A01G427200*, which encodes a cold-responsive protein, was highly upregulated in both mutant lines, suggesting that phytochromes play an important role in suppressing the cold tolerance pathway in wheat. Moreover, Konstantinov et al. [65] reported that *TraesCS2D01G425200* was significantly upregulated in wheat varieties with different drought tolerance under cold stress. Furthermore, cold acclimation increased the expression of cold-responsive proteins in both DM1 and CS; however, DM1 had a higher level of gene expression than CS (Table 2).

## 5. Conclusions

In summary, DM1 exhibited a higher cold resistance than CS by increasing the activities of ROS-scavenging enzymes, including POD, CAT, and APX, to manage cold stress-induced secretion of ROS. Consistent with the enzyme levels, transcriptome analysis showed that cold acclimation significantly increased the expression of POD and APX genes in DM1 compared with CS. Additionally, cold acclimation increased the expression of eight CAPs, six COR proteins, and 10 LEA proteins in both varieties; however, DM1 had higher expression levels of the proteins. Overall, these findings showed that DM1 exhibited a higher cold tolerance than CS by increasing the expression of ROS-scavenging and cold tolerance genes, which increased the antioxidant capacity of the plant and suppressed cold-induced cell damage.

**Supplementary Materials:** The following supporting information can be downloaded at: https://www.mdpi.com/article/10.3390/agronomy13020605/s1, Figure S1: Transcriptome profiling of CS and DM1; Figure S2: Histogram of CS and DM1 high expression gene stacking under different treatments; Figure S3: Comparison of RNA-seq results and qRT-PCR analysis of gene expression; Figure S4: FPKM values of differentially expressed POD among wheat varieties under different temperature treatments; Figure S5: FPKM values of differentially expressed CBF among wheat varieties under different temperature treatments; Figure S6: FPKM values of differentially expressed ICE among wheat varieties under different temperature treatments; Figure S7: FPKM values of differentially expressed cold shock protein among wheat varieties under different temperature treatments; Figure S8: FPKM values of differentially expressed Cold acclimation protein genes among wheat varieties under different temperature treatments; Figure S9: FPKM values of differentially expressed Cold responsive protein gene among wheat varieties under different temperature treatments; Figure S10: FPKM values of differentially expressed Late embryogenesis abundant protein among wheat varieties under different temperature treatments; Table S1: Differentially expressed genes between CS and DM1; Table S2: The FPKM of ROS-scavenging DEGs between CS and DM1; Table S3: The FPKM of cold protein DEGs between CS and DM1.

**Author Contributions:** Conceptualization, X.L. (Xiaoguang Lu) and X.W.; formal analysis, X.L. (Xiaoguang Lu), Y.W., C.T., C.L., Y.D. and X.L. (Xin Liu); validation, X.L. (Xiaoguang Lu), C.L., Y.D. and N.L. data curation, Y.W. and C.T.; investigation, Y.W., C.T. and L.F.; resources, J.L.; writing—original draft preparation, X.W.; writing—review and editing, X.W.; funding acquisition, X.W. All authors have read and agreed to the published version of the manuscript.

**Funding:** This work was supported by the National Natural Science Foundation of China (32001453); by the Academic Backbone Program of Northeast Agricultural University (20XG02); by Collaborative Innovation and Extension System of modern Agricultural Industrial Technology Wheat in Heilongjiang Province; and by Scientific Observation and Experiment Station of Crop Cultivation in

Northeast China of Ministry of Agriculture and Rural Affairs/Key Open Laboratory of Crop Variety Improvement and Physiological Ecology in Cold Region.

**Institutional Review Board Statement:** Not applicable.

**Informed Consent Statement:** Not applicable.

**Data Availability Statement:** Not applicable.

**Acknowledgments:** We thank Ruixu Biotechnology Company for their help with experiments related to high-throughput sequencing.

**Conflicts of Interest:** The authors declare no conflict of interest.

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
