# Peer review of "Physiological and Transcriptome Analysis Reveals the Differences in Genes of Antioxidative Defense Components and Cold-Related Proteins in Winter and Spring Wheat during Cold Acclimation"

_agronomy, doi:10.3390/agronomy13020605_

Round 1
Reviewer 1 Report
The research performed by Lu et. al. are very much helpful in describing tolerance mechanism in plants under cold stress. Authors explored the level of ROS produced by plant under cold stress and discuss how the enzymatic (SOD, CAT, APX, POD) and non-enzymatic (AsA, GSH) ROS scavengers gene expressions play a prime role in combating cold tolerance in wheat varieties. Still there are several grammatical and syntax errors which are mandatory to remove before its final publication. Some major suggestions are as noted below are;
Authors are suggested to edit the title as Physiological and transcriptome analysis reveals the differences in genes of antioxidative defense components and cold-related proteins in winter and spring wheat during cold acclimation or write a suitable topic
Please use AsA for denoting Ascorbic acid or ascorbate throughout the manuscript.
Abstract should be more focused and goal oriented at present stage abstract is very poor. Please rewrite the abstract and focused on finding of study in last paragraph.
Avoid the use of abbreviations in abstract section
Introduction
1. Introduction needs some up gradation therefore authors are suggested to include some recent references. Please discuss why the author choose wheat as crop for study cold tolerance in last paragraph of introduction section
2. In last paragraph the sentence especially the role the antioxidant enzyme system can be replaced by especially the role of non-enzymatic and enzymatic antioxidants.
3. Please use italics for writing name of plants like Triticum aestivum or any others
4. Please use the word activity or activities for enzymes (SOD, CAT, POD, APX, and GPX) and content or level for non-enzymatic components (AsA, GSH) throughout the manuscript in all section where these components comes
Material and methods
1. Define the conditions of treatment.
2. Determination of physiological indices can be replaced by Determination of electrical conductivity, ROS content and antioxidants
Rest part is well written
Results
3.1 In last CS had significantly higher conductivity than DM1 on November 20. Please write the significant percentage increase or decrease.
Figure 1 please use relative electrical conductivity in place of relative conductivity in both manuscript and graph.
3.2 The sentence particularly, the activities of SOD, POD, CAT, APX, ASA, and GSH were evaluated can be written as particularly, the activities of SOD, POD, CAT, APX, and content of ASA, and GSH were evaluated.
As you know the SOD, APX GPX , POD are comes under enzymatic categories and AsA, GSH comes under non-enzymatic categories. Please use activities word for enzymes and content or level for AsA and GSH throughout the manuscript
In last paragraph.. the activities of ASA and GSH was significantly higher in CS than DM1 under normal condition; however, the two reactive oxygen scavengers were significantly lower in CS than in DM1 after 10 and 30 d cold acclimation . Please elaborate it .
Please write ASA as AsA throughout the manuscript.
Figure 2 Legend is not proper reflecting the study. Kindly write it in a proper scientific way.
Figure 3 Legend is not proper reflecting the study. Kindly write it in a proper scientific way
3.4 Expression analysis of reactive oxygen scavenging enzyme genes in DM1 and CS can be replaced by Expression analysis of antioxidants genes in DM1 and CS.
Please write 82248 DEGs as A total of 82, 248 DEGs
Atotal can be correct as A total
Visibility of figure 7 is not clear. Please rectify it. Legend is okay
Discussion
Subsections 4.1 and 4.2 are well written.
Use proper space in writing sentences.
Please revise whole review oncefor literary presentation
Please better explain all figures in their figure captions
Please use proper notation for hydrogen peroxide (H2O2) and superoxide anion (O2Ë™¯) throughout the manuscript.
Please add recent finding in relation to your research in discussion section and reference section.
Author Response
Response to Reviewer 1 Comments
Point 1: The research performed by Lu et. al. are very much helpful in describing tolerance mechanism in plants under cold stress. Authors explored the level of ROS produced by plant under cold stress and discuss how the enzymatic (SOD, CAT, APX, POD) and non-enzymatic (AsA, GSH) ROS scavengers gene expressions play a prime role in combating cold tolerance in wheat varieties. Still there are several grammatical and syntax errors which are mandatory to remove before its final publication. Some major suggestions are as noted below are;
Authors are suggested to edit the title as Physiological and transcriptome analysis reveals the differences in genes of antioxidative defense components and cold-related proteins in winter and spring wheat during cold acclimation or write a suitable topic
Response 1: Thank you for your suggestion. According to your suggestion, I have changed the title of the article to: “Physiological and transcriptome analysis reveals the differences in genes of antioxidative defense components and cold-related proteins in winter and spring wheat during cold acclimation ”
Point 2: Please use AsA for denoting Ascorbic acid or ascorbate throughout the manuscript
Response 2: Thank you very much for pointing out this error in my article.I have changed the ASA in the manuscript to AsA.
Point 3: Abstract should be more focused and goal oriented at present stage abstract is very poor. Please rewrite the abstract and focused on finding of study in last paragraph.
Response 3: Thank you for your suggestion. According to your suggestion, I have rewrited the abstract:
Recent findings suggest that cold acclimation can enhance cold resistance in wheat. Dongnongdongmai 1 (DM1) is a winter wheat variety that can overwinter at -30 °C; however, its cold acclimation mechanism is yet to be fully elucidated. Here, we elucidated the potential mechanisms of cold acclimation in DM1 and China Spring (CS) variety, especially the role of antioxidant system, using transcriptome and physiological analyses. Cold stress increased H2O2 and O2- production in both varieties; however, CS had higher contents of H2O2 and O2- than DM1. Moreover, cold significantly increased ROS scavenging activities in DM1, especially at 30 d after exposure. Gene ontology (GO) analysis showed that differentially expressed peroxidase (POD) genes were enriched in antioxidant activity, with most POD genes significantly upregulated in DM1 under cold acclimation. Additionally, cold acclimation increased the expression of cold acclimation protein (CAP), late embryogenesis abundant protein (LEA), and cold responsive genes in both varieties, with higher expression levels in DM1. Overall, the results showedwe speculated that DM1 exhibited a higher cold tolerance than CS during cold acclimation by increasing the expression of POD genes, LEA, CAP, and cold responsive proteins, improving the understanding of the mechanism of cold resistance in DM1.
Point 4: Avoid the use of abbreviations in abstract section
Response 4: Thank you for your advice. Because the publication's requirement on the abstract of manuscript, which should not exceed 200 words, I have to use abbreviations. In fact, all the abbreviations in my abstract have full spelling before them.
Point 5: Introduction
(1)Introduction needs some up gradation therefore authors are suggested to include some recent references. Please discuss why the author choose wheat as crop for study cold tolerance in last paragraph of introduction section
Response 5: Thank you for your suggestion. According to your suggestion, I have added to introduction 6 articles published in 2020-2022 on cold resistance of wheat ( [22], [23], [29], [30], [33], [34]).
I also discussed why choose wheat as crop for study cold tolerance in introduction section:Wheat (Triticum aestivum L.) is a highly consumed staple crop worldwide, and plays an important role in food security [22]. In the past decades, cold stress during spring has been reported to cause severe losses to wheat production and grain quality [23].
Point 6:(2)In last paragraph the sentence especially the role the antioxidant enzyme system can be replaced by especially the role of non-enzymatic and enzymatic antioxidants.
Response 6: Thank you for your suggestion. According to your suggestion, I have changed.
Point 7:(3)Please use italics for writing name of plants like Triticum aestivum or any others
Response 7: Thank you for your suggestion. According to your suggestion, I use italics for species.
Point 8:(4)Please use the word activity or activities for enzymes (SOD, CAT, POD, APX, and GPX) and content or level for non-enzymatic components (AsA, GSH) throughout the manuscript in all section where these components comes
Response 8: Thank you for your suggestion. According to your suggestion, I have changed.
Point 9: Material and methods
(1)Define the conditions of treatment.
Response 9: Thank you for your suggestion. According to your suggestion,I have stated the conditions of treatment. For the modified part, please refer to 2.2.2
Point 10: (2)Determination of physiological indices can be replaced by Determination of electrical conductivity, ROS content and antioxidants
Response 10: Thank you for your suggestion. According to your suggestion, I have changed. For the modified part, please refer to 2.3
Point 11: Results
(1)3.1 In last CS had significantly higher conductivity than DM1 on November 20. Please write the significant percentage increase or decrease
Response 11: Thank you for your suggestion.According to your suggestion, I have revised the original text:
The relative electrical conductivity of CS and DM1 were determined on October 10 (before freezing) and November 20 (5 d after freezing) in 2018 and 2019. There was no significant difference in conductivity between DM1 and CS on October 10 of both years; however, CS had significantly higher conductivity than DM1 on November 20 (Figure 1C). In 2018, CS is 23.58% higher than DM1 and in 2019 CS is 31.45% higher than DM1.
Point 12:(2) Figure 1 please use relative electrical conductivity in place of relative conductivity in both manuscript and graph.
Response 12: Thank you for your suggestion.According to your suggestion, I have revised both manuscript and graph. For the modified part, please refer to Fig 1-C
Point 13:(3)3.2 The sentence particularly, the activities of SOD, POD, CAT, APX, ASA, and GSH were evaluated can be written as particularly, the activities of SOD, POD, CAT, APX, and content of ASA, and GSH were evaluated.
Response 13: Thank you for your suggestion. According to your suggestion, I have rewrited the sentence.
Point 14:(4)As you know the SOD, APX GPX , POD are comes under enzymatic categories and AsA, GSH comes under non-enzymatic categories. Please use activities word for enzymes and content or level for AsA and GSH throughout the manuscript
Response 14: Thank you for your suggestion.According to your suggestion, I have revised.
Point 15:(5) In last paragraph.. the activities of ASA and GSH was significantly higher in CS than DM1 under normal condition; however, the two reactive oxygen scavengers were significantly lower in CS than in DM1 after 10 and 30 d cold acclimation . Please elaborate it.
Response 15: Thank you for your suggestion. According to your suggestion, I have changed:The contents of AsA and GSH was significantly higher in DM1 (AsA:1211.45 μg.g-1FW and GSH: 10.55 mg.g-1FW ) than in CS (AsA: 987.16 μg.g-1FW and GSH:8.33 mg.g-1FW) after 30d of cold acclimation (Figure 2G–H). These results indicated that low temperature acclimation stimulated the synthesis of cold resistant varieties DM1 AsA and GSH.
Point 16:(6)Please write ASA as AsA throughout the manuscript.
Response 16: Thank you for your suggestion. According to your suggestion, I have changed.
Point 17:(7)Figure 2 Legend is not proper reflecting the study. Kindly write it in a proper scientific way.
Response 17: Thank you for your advice, I have revised. Please refer to legend of Figure 2 in manuscript
Point 18:(8)Figure 3 Legend is not proper reflecting the study. Kindly write it in a proper scientific way
Response 18: Thank you for your suggestion. In order to explain the consistency of RNA-seq and qRT-PCR, I added regression analysis, For details, please see the new Figure 3. The original Figure 3 was used as the Figure S3, and legend descriptions were supplemented. For details, please see the legend of Figure S3
Point 19:(9)3.4 Expression analysis of reactive oxygen scavenging enzyme genes in DM1 and CS can be replaced by Expression analysis of antioxidants genes in DM1 and CS.
Response 19: Thank you for your suggestion. According to your suggestion, I have rewrited the sentence.
Point 20:(10)Please write 82248 DEGs as A total of 82, 248 DEGs
Response 20: Thank you for your suggestion. According to your suggestion, I have changed 82248 DEGs to 82, 248 DEGs
Point 21:(11)Atotal can be correct as A total
Response 21: Thank you for your suggestion. According to your suggestion, I have changed Atotal to A total
Point 22:(12) Visibility of figure 7 is not clear. Please rectify it. Legend is okay
Response 22: Thank you for your suggestion. According to your suggestion, I have changed
Point 23: Discussion
(1)Subsections 4.1 and 4.2 are well written.Use proper space in writing sentences.
Response 23: Thank you for your suggestion. According to your suggestion, I have changed.
Point 24:(2)Please revise whole review once for literary presentation
Response 24: Thank you for your suggestion. According to your suggestion, I have changed.
Point 25:(3)Please better explain all figures in their figure captions
Response 25: Thank you for your suggestion. According to your suggestion, I have explained all figures captions .
Point 26:(4)Please use proper notation for hydrogen peroxide (H2O2) and superoxide anion (O2Ë™¯) throughout the manuscript.
Response 26: Thank you for your suggestion. According to your suggestion, I have use proper notation for hydrogen peroxide (H2O2) and superoxide anion (O2Ë™¯) throughout the manuscript.
Point 27:(5)Please add recent finding in relation to your research in discussion section and reference section.
Response 27: Thank you for your suggestion. According to your suggestion, I have added the latest findings related to the research in the discussion section and the reference section.

Reviewer 2 Report
I really appreciate the efforts of authors
background section need minor improvement
As the authors are not English native, English language need minor revision
Author Response
Response to Reviewer 2 Comments
I really appreciate the efforts of authors
background section need minor improvement
As the authors are not English native, English language need minor revision
Response 1: Thank you very much for your recognition and advice. According to your suggestion, I have added to introduction 6 articles published in 2020-2022 on cold resistance of wheat ( [22], [23], [29], [30], [33], [34]).
Some of these articles are about why I chose wheat as the research object; some are about the research progress of cold-resistant variety DM1, which is also the research material I chose in this manuscript; and some are about the progress of identification of resistance genes by transcriptome screening. All these supplementary literatures are well related to the research in this paper.
Response 2: Thank you very much for your suggestion. “English language need minor revision”. I submitted the manuscript to a professional language editing revision agency for completion. Please see the certificate attached.

Reviewer 3 Report
Introduction:
1. In a paragraph, explain the significance of transcriptome studies and provide references from earlier research about cold stress in wheat.
2. Line: However, the role of the antioxidant system, especially ROS-scavenging enzymes associated genes in acclimation in wheat is yet to examined. This statement is not very clear?
Results
1.Provide a detailed explanation of the Figure legend along with a detailed explanation of the statics used
2. Figure 2 can be shown as a line plot, making the data more visually appealing and understandable. Also, Blue color is for ‘CS’, while ‘Orange’ is for DM. Make the legend with the color code and on the X axis start writing only 1d/24°C, 10d/10°C, 10d/24°C, 30d/10°C, 30d/24°C ; see example
1. What is the gene ontology status of 8,576 and 3,584 TUs were specific to CS and DM1, respectively. I think it’s important to give some information about them.
2. In Figure 3, where the authors compare RNA seq data with qRT-PCR, regression-based analysis can be used to make the comparison rather than just drawing a bar graph. Please make improvements and include r2 values for each gene. Again, the figure legend is not instructive; instead, be extremely clear about what the color code means so that everyone can comprehend it.
3. Clearly state the DEG for each paired condition, plot the up- and down-DEGs in a stacked plot, then display a venn diagram to clarify the shared/unique DEG, and finally, decide which set combination to use for a more thorough analysis.
4. Figure 4 (C-J). The y axis (log2FC) can range from 4 to -4, making the variance in gene expression obvious.
Author Response
Response to Reviewer 3 Comments
Point 1:Introduction
(1)In a paragraph, explain the significance of transcriptome studies and provide references from earlier research about cold stress in wheat.
Response 1: Thank you for your advice.According to your suggestion, I have added references in paragraph :
Omics technology offers opportunity to study the regulatory mechanisms of complex networks related to cold hardiness. Lv et al [33] performed a transcriptome analysis of wild-type wheat (Jimai325) and cold sensitive mutants and identified eight upregulated enzymes in Jimai325, all of which were involved in sucrose and amino acid biosynthesis pathway. Additionally, Tian et al [34] reported that differentially expressed genes (DEGs) in DM1 exposed to temperatures ranging from -25 to 0 °C were involved in oxidation-reduction, protein phosphorylation, and carbohydrate metabolism.
Point 2: (2)Line: However, the role of the antioxidant system, especially ROS-scavenging enzymes associated genes in acclimation in wheat is yet to examined. This statement is not very clear?
Response 2: Thank you for your suggestion. According to your suggestion, I have modified :
However, studies are yet to elucidate the role of the antioxidant system in cold tolerance in DM1, especially enzymes and genes involved in ROS scavenging under cold stress.
Point 3: Results
(1) Provide a detailed explanation of the Figure legend along with a detailed explanation of the statics used
Response 3: Thank you for your suggestion. According to your suggestion, I have detailed explanation legend of the Figure 1,Figure 2, Figure3
Point 4: (2) Figure 2 can be shown as a line plot, making the data more visually appealing and understandable. Also, Blue color is for ‘CS’, while ‘Orange’ is for DM. Make the legend with the color code and on the X axis start writing only 1d/24°C, 10d/10°C, 10d/24°C, 30d/10°C, 30d/24°C ; see example
Response 4: Thank you for your suggestion. According to your suggestion, I have changed the image to a line plot.
Point 5: (3) What is the gene ontology status of 8,576 and 3,584 TUs were specific to CS and DM1, respectively. I think it’s important to give some information about them
Response 5: Thank you for your suggestion. According to your suggestion I added the following: Additionally, 8,576 and 3,584 TUs were specific to CS and DM1, respectively, and 62,241 TUs were detected in both genotypes (Figure S1-B). The pathways in which DM1-specific expressed genes are involved are mostly developmental and response to stimulus (Figure S1-C).
Point 6: (3) In Figure 3, where the authors compare RNA seq data with qRT-PCR, regression-based analysis can be used to make the comparison rather than just drawing a bar graph. Please make improvements and include r2 values for each gene. Again, the figure legend is not instructive; instead, be extremely clear about what the color code means so that everyone can comprehend it.
Response 6: Thank you for your suggestion. In order to explain the consistency of RNA-seq and qRT-PCR, I added regression analysis new Figure 3. The original Figure 3 was used as the Figure S3, and legend descriptions were supplemented. For details, please see the legend of Figure 3 and Figure S3
Point 7: (4)Clearly state the DEG for each paired condition, plot the up and down DEGs in a stacked plot, then display a venn diagram to clarify the shared unique DEG, and finally, decide which set combination to use for a more thorough analysis.
Response 7: Thank you for your advice. According to your suggestion, I added Figure S2
Point 8: (5)Figure 4 (C-J). The y axis (log2FC) can range from 4 to -4, making the variance in gene expression obvious.
Response 8: Thank you very much for pointing out this deficiency in my manuscript, I have revised the y axis (log2FC) can range from 4 to -4

Round 2
Reviewer 1 Report
The authors have addressed all the comments properly.
Reviewer 3 Report
The authors have made numerous corrections, and the manuscript is now significantly enhanced. I would still recommend proofreading for proper citation, grammar, and English usage. I strongly endorse this manuscript for publication in this journal.